# Optimal Ridge Detection using Coverage Risk

**Yen-Chi Chen**
Department of Statistics
Carnegie Mellon University
yenchic@andrew.cmu.edu

**Christopher R. Genovese**
Department of Statistics
Carnegie Mellon University
genovese@stat.cmu.edu

**Shirley Ho**
Department of Physics
Carnegie Mellon University
shirleyh@andrew.cmu.edu

**Larry Wasserman**
Department of Statistics
Carnegie Mellon University
larry@stat.cmu.edu

## Abstract

We introduce the concept of coverage risk as an error measure for density ridge estimation. The coverage risk generalizes the mean integrated square error to set estimation. We propose two risk estimators for the coverage risk and we show that we can select tuning parameters by minimizing the estimated risk. We study the rate of convergence for coverage risk and prove consistency of the risk estimators. We apply our method to three simulated datasets and to cosmology data. In all the examples, the proposed method successfully recover the underlying density structure.

## 1 Introduction

Density ridges [10, 22, 15, 6] are one-dimensional curve-like structures that characterize high density regions. Density ridges have been applied to computer vision [2], remote sensing [21], biomedical imaging [1], and cosmology [5, 7]. Density ridges are similar to the principal curves [17, 18, 27]. Figure 1 provides an example for applying density ridges to learn the structure of our Universe.

To detect density ridges from data, [22] proposed the 'Subspace Constrained Mean Shift (SCMS)' algorithm. SCMS is a modification of usual mean shift algorithm [14, 8] to adapt to the local geometry. Unlike mean shift that pushes every mesh point to a local mode, SCMS moves the meshes along a projected gradient until arriving at nearby ridges. Essentially, the SCMS algorithm detects the ridges of the kernel density estimator (KDE). Therefore, the SCMS algorithm requires a pre-selected parameter $h$, which acts as the role of smoothing bandwidth in the kernel density estimator.

Despite the wide application of the SCMS algorithm, the choice of $h$ remains an unsolved problem. Similar to the density estimation problem, a poor choice of $h$ results in over-smoothing or under-smoothing for the density ridges. See the second row of Figure 1.

In this paper, we introduce the concept of coverage risk which is a generalization of the mean integrated expected error from function estimation. We then show that one can consistently estimate the coverage risk by using data splitting or the smoothed bootstrap. This leads us to a data-driven selection rule for choosing the parameter $h$ for the SCMS algorithm. We apply the proposed method to several famous datasets including the spiral dataset, the three spirals dataset, and the NIPS dataset. In all simulations, our selection rule allows the SCMS algorithm to detect the underlying structure of the data.

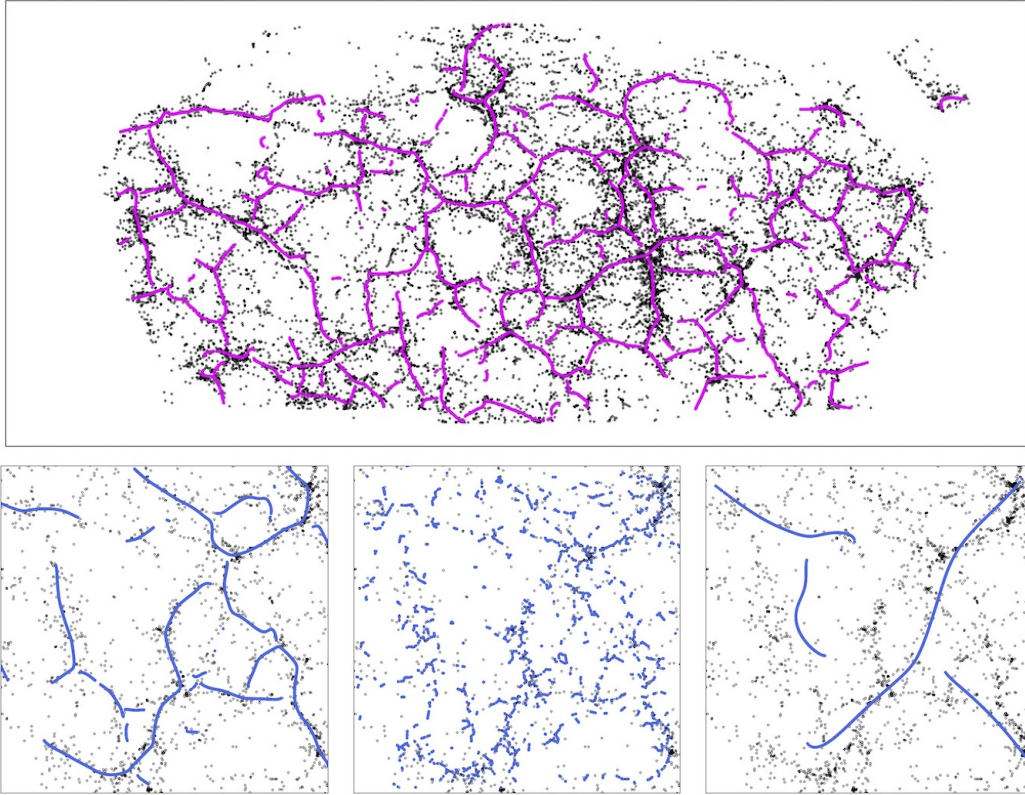

Figure 1: The cosmic web. This is a slice of the observed Universe from the Sloan Digital Sky Survey. We apply the density ridge method to detect filaments [7]. The top row is one example for the detected filaments. The bottom row shows the effect of smoothing. Bottom-Left: optimal smoothing. Bottom-Middle: under-smoothing. Bottom-Right: over-smoothing. Under optimal smoothing, we detect an intricate filament network. If we under-smooth or over-smooth the dataset, we cannot find the structure.

## 1.1   Density Ridges

Density ridges are defined as follows. Assume $X_1, \cdots, X_n$ are independently and identically distributed from a smooth probability density function $p$ with compact support $\mathbb{K}$. The density ridges [10, 15, 6] are defined as

$$R = \{x \in \mathbb{K} : V(x)V(x)^T \nabla p(x) = 0, \lambda_2(x) < 0\},$$

where $V(x) = [v_2(x), \cdots v_d(x)]$ with $v_j(x)$ being the eigenvector associated with the ordered eigenvalue $\lambda_j(x)$ ($\lambda_1(x) \geq \cdots \geq \lambda_d(x)$) for Hessian matrix $H(x) = \nabla\nabla p(x)$. That is, $R$ is the collection of points whose *projected gradient* $V(x)V(x)^T \nabla p(x) = 0$. It can be shown that (under appropriate conditions), $R$ is a collection of 1-dimensional smooth curves (1-dimensional manifolds) in $\mathbb{R}^d$.

The SCMS algorithm is a plug-in estimate for $R$ by using

$$\widehat{R}_n = \left\{ x \in \mathbb{K} : \widehat{V}_n(x)\widehat{V}_n(x)^T \nabla \widehat{p}_n(x) = 0, \widehat{\lambda}_2(x) < 0 \right\},$$

where $\widehat{p}_n(x) = \frac{1}{nh^d} \sum_{i=1}^n K\left(\frac{x-X_i}{h}\right)$ is the KDE and $\widehat{V}_n$ and $\widehat{\lambda}_2$ are the associated quantities defined by $\widehat{p}_n$. Hence, one can clearly see that the parameter $h$ in the SCMS algorithm plays the same role of smoothing bandwidth for the KDE.

## 2 Coverage Risk

Before we introduce the coverage risk, we first define some geometric concepts. Let $\mu_\ell$ be the $\ell$-dimensional Hausdorff measure [13]. Namely, $\mu_1(A)$ is the length of set $A$ and $\mu_2(A)$ is the area of $A$. Let $d(x, A)$ be the projection distance from point $x$ to a set $A$. We define $U_R$ and $U_{\widehat{R}_n}$ as random variables uniformly distributed over the true density ridges $R$ and the ridge estimator $\widehat{R}_n$ respectively. Assuming $R$ and $\widehat{R}_n$ are given, we define the following two random variables

$$W_n = d(U_R, \widehat{R}_n), \quad \widetilde{W}_n = d(U_{\widehat{R}_n}, R). \tag{1}$$

Note that $U_R, U_{\widehat{R}_n}$ are random variables while $R, \widehat{R}_n$ are sets. $W_n$ is the distance from a randomly selected point on $R$ to the estimator $\widehat{R}_n$ and $\widetilde{W}_n$ is the distance from a random point on $\widehat{R}_n$ to $R$.

Let $\mathsf{Haus}(A, B) = \inf\{r : A \subset B \oplus r, B \subset A \oplus r\}$ be the Hausdorff distance between $A$ and $B$ where $A \oplus r = \{x : d(x, A) \leq r\}$. The following lemma gives some useful properties about $W_n$ and $\widetilde{W}_n$.

**Lemma 1** *Both random variables $W_n$ and $\widetilde{W}_n$ are bounded by* $\mathsf{Haus}(\widehat{M}_n, M)$. *Namely,*

$$0 \leq W_n \leq \mathsf{Haus}(\widehat{R}_n, R), \quad 0 \leq \widetilde{W}_n \leq \mathsf{Haus}(\widehat{R}_n, R). \tag{2}$$

*The cumulative distribution function (CDF) for $W_n$ and $\widetilde{W}_n$ are*

$$\mathbb{P}(W_n \leq r | \widehat{R}_n) = \frac{\mu_1\left(R \cap (\widehat{R}_n \oplus r)\right)}{\mu_1(R)}, \quad \mathbb{P}(\widetilde{W}_n \leq r | \widehat{R}_n) = \frac{\mu_1\left(\widehat{R}_n \cap (R \oplus r)\right)}{\mu_1\left(\widehat{R}_n\right)}. \tag{3}$$

*Thus, $\mathbb{P}(W_n \leq r | \widehat{R}_n)$ is the ratio of $R$ being covered by padding the regions around $\widehat{R}_n$ at distance $r$.*

This lemma follows trivially by definition so that we omit its proof. Lemma 1 links the random variables $W_n$ and $\widetilde{W}_n$ to the Hausdorff distance and the coverage for $R$ and $\widehat{R}_n$. Thus, we call them *coverage* random variables. Now we define the $\mathcal{L}_1$ and $\mathcal{L}_2$ *coverage risk* for estimating $R$ by $\widehat{R}_n$ as

$$\mathsf{Risk}_{1,n} = \frac{\mathbb{E}(W_n + \widetilde{W}_n)}{2}, \quad \mathsf{Risk}_{2,n} = \frac{\mathbb{E}(W_n^2 + \widetilde{W}_n^2)}{2}. \tag{4}$$

That is, $\mathsf{Risk}_{1,n}$ (and $\mathsf{Risk}_{2,n}$) is the expected (square) projected distance between $R$ and $\widehat{R}_n$. Note that the expectation in (4) applies to both $\widehat{R}_n$ and $U_R$. One can view $\mathsf{Risk}_{2,n}$ as a generalized mean integrated square errors (MISE) for sets.

A nice property of $\mathsf{Risk}_{1,n}$ and $\mathsf{Risk}_{2,n}$ is that they are not sensitive to outliers of $R$ in the sense that a small perturbation of $R$ will not change the risk too much. On the contrary, the Hausdorff distance is very sensitive to outliers.

### 2.1 Selection for Tuning Parameters Based on Risk Minimization

In this section, we will show how to choose $h$ by minimizing an estimate of the risk.

We propose two risk estimators. The first estimator is based on the *smoothed bootstrap* [25]. We sample $X_1^*, \cdots X_n^*$ from the KDE $\widehat{p}_n$ and recompute the estimator $\widehat{R}_n^*$. The we estimate the risk by

$$\widehat{\mathsf{Risk}}_{1,n} = \frac{\mathbb{E}(W_n^* + \widetilde{W}_n^* | X_1, \cdots, X_n)}{2}, \quad \widehat{\mathsf{Risk}}_{2,n} = \frac{\mathbb{E}(W_n^{*2} + \widetilde{W}_n^{*2} | X_1, \cdots, X_n)}{2}, \tag{5}$$

where $W_n^* = d(U_{\widehat{R}_n}, \widehat{R}_n^*)$ and $\widetilde{W}_n^* = d(U_{\widehat{R}_n^*}, \widehat{R}_n)$.

The second approach is to use data splitting. We randomly split the data into $X_{11}^\dagger, \cdots, X_{1m}^\dagger$ and $X_{21}^\dagger, \cdots, X_{2m}^\dagger$, assuming $n$ is even and $2m = n$. We compute the estimated manifolds by using half of the data, which we denote as $\widehat{R}_{1,n}^\dagger$ and $\widehat{R}_{2,n}^\dagger$. Then we compute

$$\widehat{\mathsf{Risk}}_{1,n}^\dagger = \frac{\mathbb{E}(W_{1,n}^\dagger + W_{2,n}^\dagger | X_1, \cdots, X_n)}{2}, \quad \widehat{\mathsf{Risk}}_{2,n}^\dagger = \frac{\mathbb{E}(W_{1,n}^{\dagger 2} + W_{2,n}^{\dagger 2} | X_1, \cdots, X_n)}{2}, \quad (6)$$

where $W_{1,n}^\dagger = d(U_{\widehat{R}_{1,n}^\dagger}, \widehat{R}_{2,n}^\dagger)$ and $W_{2,n}^\dagger = d(U_{\widehat{R}_{2,n}^\dagger}, \widehat{R}_{1,n}^\dagger)$.

Having estimated the risk, we select $h$ by

$$h^* = \underset{h \leq \bar{h}_n}{\mathrm{argmin}} \ \widehat{\mathsf{Risk}}_{1,n}^\dagger, \quad (7)$$

where $\bar{h}_n$ is an upper bound by the normal reference rule [26] (which is known to oversmooth, so that we only select $h$ below this rule). Moreover, one can choose $h$ by minimizing $\mathcal{L}_2$ risk as well.

In [11], they consider selecting the smoothing bandwidth for local principal curves by self-coverage. This criterion is a different from ours. The self-coverage counts data points. The self-coverage is a monotonic increasing function and they propose to select the bandwidth such that the derivative is highest. Our coverage risk yields a simple trade-off curve and one can easily pick the optimal bandwidth by minimizing the estimated risk.

## 3 Manifold Comparison by Coverage

The concepts of coverage in previous section can be generalized to investigate the difference between two manifolds. Let $M_1$ and $M_2$ be an $\ell_1$-dimensional and an $\ell_2$-dimensional manifolds ($\ell_1$ and $\ell_2$ are not necessarily the same). We define the coverage random variables

$$W_{12} = d(U_{M_1}, M_2), \quad W_{21} = d(U_{M_2}, M_1). \quad (8)$$

Then by Lemma 1, the CDF for $W_{12}$ and $W_{21}$ contains information about how $M_1$ and $M_2$ are different from each other:

$$\mathbb{P}(W_{12} \leq r) = \frac{\mu_{\ell_1}(M_1 \cap (M_2 \oplus r))}{\mu_{\ell_2}(M_1)}, \quad \mathbb{P}(W_{21} \leq r) = \frac{\mu_{\ell_2}(M_2 \cap (M_1 \oplus r))}{\mu_{r_2}(M_1)}. \quad (9)$$

$\mathbb{P}(W_{12} \leq r)$ is the coverage on $M_1$ by padding regions with distance $r$ around $M_2$.

We call the plots of the CDF of $W_{12}$ and $W_{21}$ *coverage diagrams* since they are linked to the coverage over $M_1$ and $M_2$. The coverage diagram allows us to study how two manifolds are different from each other. When $\ell_1 = \ell_2$, the coverage diagram can be used as a similarity measure for two manifolds. When $\ell_1 \neq \ell_2$, the coverage diagram serves as a measure for quality of representing high dimensional objects by low dimensional ones. A nice property for coverage diagram is that we can approximate the CDF for $W_{12}$ and $W_{21}$ by a mesh of points (or points uniformly distributed) over $M_1$ and $M_2$. In Figure 2 we consider a Helix dataset whose support has dimension $d = 3$ and we compare two curves, a spiral curve (green) and a straight line (orange), to represent the Helix dataset. As can be seen from the coverage diagram (right panel), the green curve has better coverage at each distance (compared to the orange curve) so that the spiral curve provides a better representation for the Helix dataset.

In addition to the coverage diagram, we can also use the following $\mathcal{L}_1$ and $\mathcal{L}_2$ losses as summary for the difference:

$$\mathsf{Loss}_1(M_1, M_2) = \frac{\mathbb{E}(W_{12} + W_{21})}{2}, \quad \mathsf{Loss}_2(M_1, M_2) = \frac{\mathbb{E}(W_{12}^2 + W_{21}^2)}{2}. \quad (10)$$

The expectation is take over $U_{M_1}$ and $U_{M_2}$ and both $M_1$ and $M_2$ here are fixed. The risks in (4) are the expected losses:

$$\mathsf{Risk}_{1,n} = \mathbb{E}\left(\mathsf{Loss}_1(\widehat{M}_n, M)\right), \quad \mathsf{Risk}_{2,n} = \mathbb{E}\left(\mathsf{Loss}_2(\widehat{M}_n, M)\right). \quad (11)$$

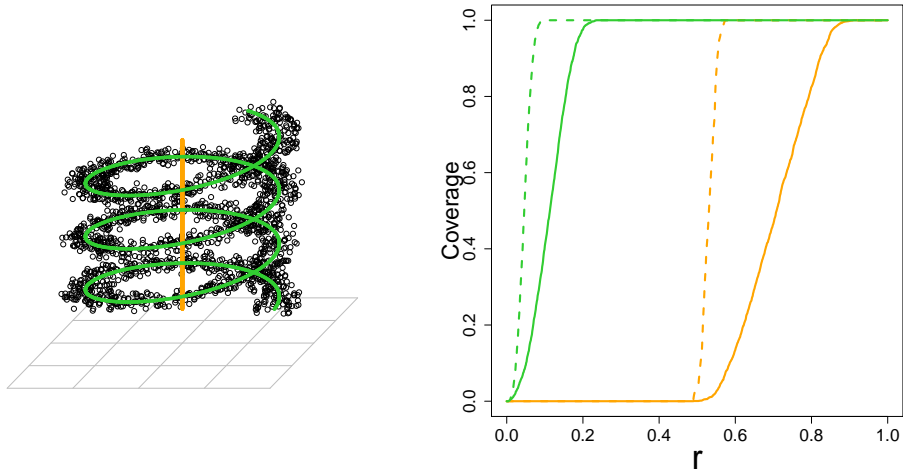

Figure 2: The Helix dataset. The original support for the Helix dataset (black dots) are a 3-dimensional regions. We can use green spiral curves ($d = 1$) to represent the regions. Note that we also provide a bad representation using a straight line (orange). The coverage plot reveals the quality for representation. Left: the original data. Dashed line is coverage from data points (black dots) over green/orange curves in the left panel and solid line is coverage from green/orange curves on data points. Right: the coverage plot for the spiral curve (green) versus a straight line (orange).

## 4 Theoretical Analysis

In this section, we analyze the asymptotic behavior for the coverage risk and prove the consistency for estimating the coverage risk by the proposed method. In particular, we derive the asymptotic properties for the density ridges. We only focus on $\mathcal{L}_2$ risk since by Jensen's inequality, the $\mathcal{L}_2$ risk can be bounded by the $\mathcal{L}_1$ risk.

Before we state our assumption, we first define the orientation of density ridges. Recall that the density ridge $R$ is a collection of one dimensional curves. Thus, for each point $x \in R$, we can associate a unit vector $e(x)$ that represent the orientation of $R$ at $x$. The explicit formula for $e(x)$ can be found in Lemma 1 of [6].

**Assumptions.**

(R) There exist $\beta_0, \beta_1, \beta_2, \delta_R > 0$ such that for all $x \in R \oplus \delta_R$,
$$\lambda_2(x) \leq -\beta_1, \quad \lambda_1(x) \geq \beta_0 - \beta_1, \quad \|\nabla p(x)\| \|p^{(3)}(x)\|_{\max} \leq \beta_0(\beta_1 - \beta_2), \quad (12)$$
where $\|p^{(3)}(x)\|_{\max}$ is the element wise norm to the third derivative. And for each $x \in R$, $|e(x)^T \nabla p(x)| \geq \frac{\lambda_1(x)}{\lambda_1(x) - \lambda_2(x)}$.

(K1) The kernel function $K$ is three times bounded differetiable and is symmetric, non-negative and
$$\int x^2 K^{(\alpha)}(x) dx < \infty, \qquad \int \left( K^{(\alpha)}(x) \right)^2 dx < \infty$$
for all $\alpha = 0, 1, 2, 3$.

(K2) The kernel function $K$ and its partial derivative satisfies condition $K_1$ in [16]. Specifically, let
$$\mathcal{K} = \left\{ y \mapsto K^{(\alpha)} \left( \frac{x - y}{h} \right) : x \in \mathbb{R}^d, h > 0, |\alpha| = 0, 1, 2 \right\} \quad (13)$$
We require that $\mathcal{K}$ satisfies
$$\sup_P N \left( \mathcal{K}, L_2(P), \epsilon \|F\|_{L_2(P)} \right) \leq \left( \frac{A}{\epsilon} \right)^v \quad (14)$$

for some positive number $A, v$, where $N(T, d, \epsilon)$ denotes the $\epsilon$-covering number of the metric space $(T, d)$ and $F$ is the envelope function of $\mathcal{K}$ and the supreme is taken over the whole $\mathbb{R}^d$. The $A$ and $v$ are usually called the VC characteristics of $\mathcal{K}$. The norm $\|F\|_{L_2(P)} = \sup_P \int |F(x)|^2 dP(x)$.

Assumption (R) appears in [6] and is very mild. The first two inequality in (12) are just the bound on eigenvalues. The last inequality requires the density around ridges to be smooth. The latter part of (R) requires the direction of ridges to be similar to the gradient direction. Assumption (K1) is the common condition for kernel density estimator see e.g. [28] and [24]. Assumption (K2) is to regularize the classes of kernel functions that is widely assumed [12, 15, 4]; any bounded kernel function with compact support satisfies this condition. Both (K1) and (K2) hold for the Gaussian kernel.

Under the above condition, we derive the rate of convergence for the $\mathcal{L}_2$ risk.

**Theorem 2** *Let* $\mathsf{Risk}_{2,n}$ *be the* $\mathcal{L}_2$ *coverage risk for estimating the density ridges and level sets. Assume (K1–2) and (R) and $p$ is at least four times bounded differentiable. Then as $n \to \infty$, $h \to 0$ and $\frac{\log n}{nh^{d+6}} \to 0$*

$$\mathsf{Risk}_{2,n} = B_R^2 h^4 + \frac{\sigma_R^2}{nh^{d+2}} + o(h^4) + o\left(\frac{1}{nh^{d+2}}\right),$$

*for some $B_R$ and $\sigma_R^2$ that depends only on the density $p$ and the kernel function $K$.*

The rate in Theorem 2 shows a bias-variance decomposition. The first term involving $h^4$ is the bias term while the latter term is the variance part. Thanks to the Jensen's inequality, the rate of convergence for $\mathcal{L}_1$ risk is the square root of the rate Theorem 2. Note that we require the smoothing parameter $h$ to decay slowly to 0 by $\frac{\log n}{nh^{d+6}} \to 0$. This constraint comes from the uniform bound for estimating third derivatives for $p$. We need this constraint since we need the smoothness for estimated ridges to converge to the smoothness for the true ridges. Similar result for density level set appears in [3, 20].

By Lemma 1, we can upper bound the $\mathcal{L}_2$ risk by expected square of the Hausdorff distance which gives the rate

$$\mathsf{Risk}_{2,n} \leq \mathbb{E}\left(\mathsf{Haus}^2(\widehat{R}_n, R)\right) = O(h^4) + O\left(\frac{\log n}{nh^{d+2}}\right) \tag{15}$$

The rate under Hausdorff distance for density ridges can be found in [6] and the rate for density ridges appears in [9]. The rate induced by Theorem 2 agrees with the bound from the Hausdorff distance and has a slightly better rate for variance (without a log-n factor). This phenomena is similar to the MISE and $\mathcal{L}_\infty$ error for nonparametric estimation for functions. The MISE converges slightly faster (by a log-n factor) than square to the $\mathcal{L}_\infty$ error.

Now we prove the consistency of the risk estimators. In particular, we prove the consistency for the smoothed bootstrap. The case of data splitting can be proved in the similar way.

**Theorem 3** *Let* $\mathsf{Risk}_{2,n}$ *be the* $\mathcal{L}_2$ *coverage risk for estimating the density ridges and level sets. Let* $\widehat{\mathsf{Risk}}_{2,n}$ *be the corresponding risk estimator by the smoothed bootstrap. Assume (K1–2) and (R) and $p$ is at least four times bounded differentiable. Then as $n \to \infty$, $h \to 0$ and $\frac{\log n}{nh^{d+6}} \to 0$,*

$$\frac{\widehat{\mathsf{Risk}}_{2,n} - \mathsf{Risk}_{2,n}}{\mathsf{Risk}_{2,n}} \xrightarrow{P} 0.$$

Theorem 3 proves the consistency for risk estimation using the smoothed bootstrap. This also leads to the consistency for data splitting.

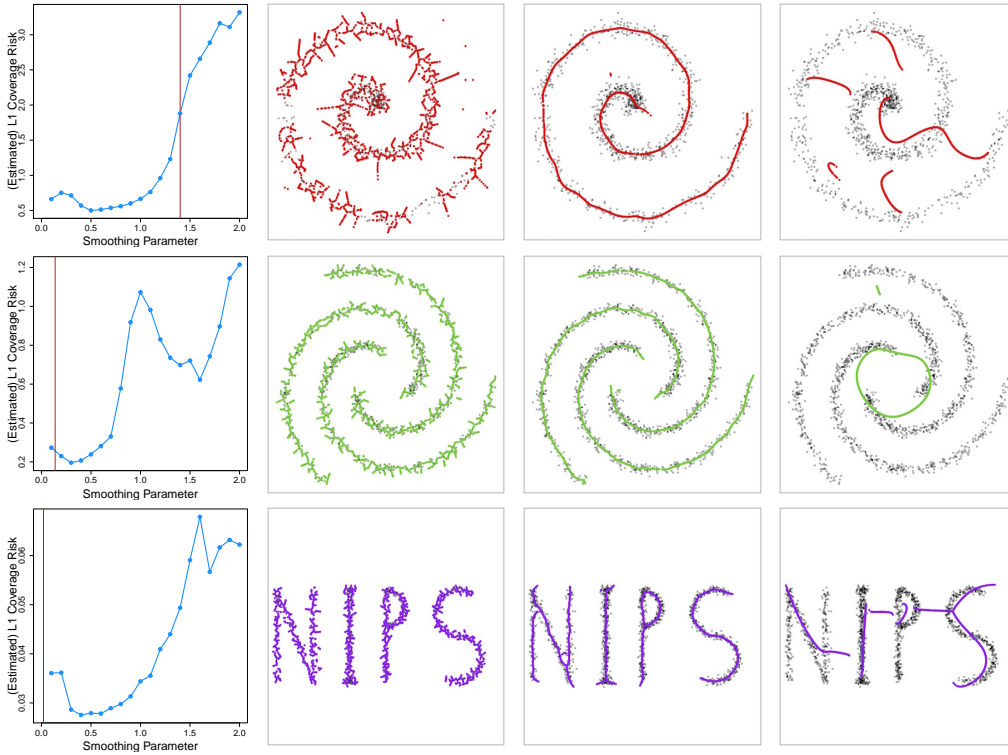

Figure 3: Three different simulation datasets. Top row: the spiral dataset. Middle row: the three spirals dataset. Bottom row: NIPS character dataset. For each row, the leftmost panel shows the estimated $\mathcal{L}_1$ coverage risk using data splitting; the red straight line indicates the bandwidth selected by least square cross validation [19], which is either undersmooth or oversmooth. Then the rest three panels, are the result using different smoothing parameters. From left to right, we show the result for under-smoothing, optimal smoothing (using the coverage risk), and over-smoothing. Note that the second minimum in the coverage risk at the three spirals dataset (middle row) corresponds to a phase transition when the estimator becomes a big circle; this is also a locally stable structure.

## 5   Applications

### 5.1   Simulation Data

We now apply the data splitting technique (7) to choose the smoothing bandwidth for density ridge estimation. Note that we use data splitting over smooth bootstrap since in practice, data splitting works better. The density ridge estimation can be done by the subspace constrain mean shift algorithm [22]. We consider three famous datasets: the spiral dataset, the three spirals dataset and a 'NIPS' dataset.

Figure 3 shows the result for the three simulation datasets. The top row is the spiral dataset; the middle row is the three spirals dataset; the bottom row is the NIPS character dataset. For each row, from left to right the first panel is the estimated $\mathcal{L}_1$ risk by using data splitting. Note that there is no practical difference between $\mathcal{L}_1$ and $\mathcal{L}_2$ risk. The second to fourth panels are under-smoothing, optimal smoothing, and over-smoothing. Note that we also remove the ridges whose density is below $0.05 \times \max_x \widehat{p}_n(x)$ since they behave like random noise. As can be seen easily, the optimal bandwidth allows the density ridges to capture the underlying structures in every dataset. On the contrary, the under-smoothing and the over-smoothing does not capture the structure and have a higher risk.

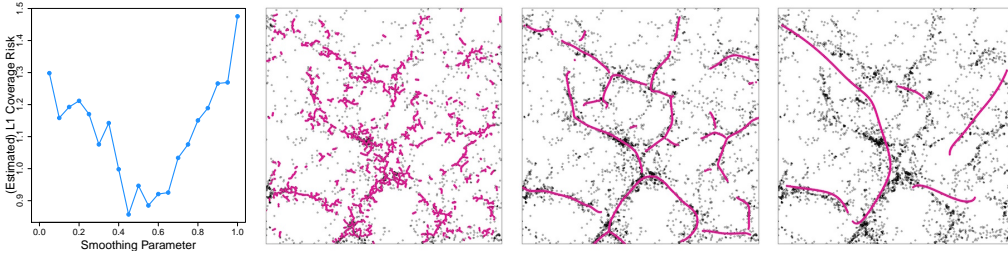

Figure 4: Another slice for the cosmic web data from the Sloan Digital Sky Survey. The leftmost panel shows the (estimated) $\mathcal{L}_1$ coverage risk (right panel) for estimating density ridges under different smoothing parameters. We estimated the $\mathcal{L}_1$ coverage risk by using data splitting. For the rest panels, from left to right, we display the case for under-smoothing, optimal smoothing, and over-smoothing. As can be seen easily, the optimal smoothing method allows the SCMS algorithm to detect the intricate cosmic network structure.

## 5.2 Cosmic Web

Now we apply our technique to the Sloan Digital Sky Survey, a huge dataset that contains millions of galaxies. In our data, each point is an observed galaxy with three features:

- z: the redshift, which is the distance from the galaxy to Earth.
- RA: the right ascension, which is the longitude of the Universe.
- dec: the declination, which is the latitude of the Universe.

These three features $(z, RA, dec)$ uniquely determine the location of a given galaxy.

To demonstrate the effectiveness of our method, we select a 2-D slice of our Universe at redshift $z = 0.050 - 0.055$ with $(RA, dec) \in [200, 240] \times [0, 40]$. Since the redshift difference is very tiny, we ignore the redshift value of the galaxies within this region and treat them as a 2-D data points. Thus, we only use $RA$ and $dec$. Then we apply the SCMS algorithm (version of [7]) with data splitting method introduced in section 2.1 to select the smoothing parameter $h$. The result is given in Figure 4. The left panel provides the estimated coverage risk at different smoothing bandwidth. The rest panels give the result for under-smoothing (second panel), optimal smoothing (third panel) and over-smoothing (right most panel). In the third panel of Figure 4, we see that the SCMS algorithm detects the filament structure in the data.

## 6 Discussion

In this paper, we propose a method using coverage risk, a generalization of mean integrated square error, to select the smoothing parameter for the density ridge estimation problem. We show that the coverage risk can be estimated using data splitting or smoothed bootstrap and we derive the statistical consistency for risk estimators. Both simulation and real data analysis show that the proposed bandwidth selector works very well in practice.

The concept of coverage risk is not limited to density ridges; instead, it can be easily generalized to other manifold learning technique. Thus, we can use data splitting to estimate the risk and use the risk estimator to select the tuning parameters. This is related to the so-called stability selection [23], which allows us to select tuning parameters even in an unsupervised learning settings.

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

—

