[Supplementary Material]

# Supplements to "Optimal Ridge Detection using Coverage Risk"

**Yen-Chi Chen**
Department of Statistics
Carnegie Mellon University
yenchic@andrew.cmu.edu

**Christopher R. Genovese**
Department of Statistics
Carnegie Mellon University
genovese@stat.cmu.edu

**Shirley Ho**
Department of Physics
Carnegie Mellon University
shirleyh@andrew.cmu.edu

**Larry Wasserman**
Department of Statistics
Carnegie Mellon University
larry@stat.cmu.edu

## 1   Proofs

Before we prove Theorem 2, we need the following lemma for comparing two curves.

**Lemma 4**  *Let $S_1, S_2$ be two bounded smooth curves in $\mathbb{R}^d$. Let $\pi_{12}: S_1 \mapsto S_2$ and $\pi_{21}: S_2 \mapsto S_1$ be the projections between them. For $a \in S_1$ and $b \in S_2$, define $g_1(a)$ and $g_2(b)$ as the unit tangent vectors for $S_1$ and $S_2$ at $a$ and $b$ respectively. Assume $S_1$ and $S_2$ are similar in the following sense:*

*(S1)  $\pi_{12}$ and $\pi_{21}$ are one-one and onto,*

*(S2)  the projections are similar:*

$$\max\left\{ \sup_{x \in S_1} \|\pi_{12}(x) - \pi_{21}^{-1}(x)\|, \sup_{x \in S_2} \|\pi_{21}(x) - \pi_{12}^{-1}(x)\| \right\} = O(\epsilon_1),$$

*(S3)  the tangent vectors are similar:*

$$\max\left\{ \sup_{x \in S_1} |g_1(x)^T g_2(\pi_{12}(x))|, \sup_{x \in S_2} |g_2(x)^T g_1(\pi_{21}(x))| \right\} = 1 + O(\epsilon_2),$$

*(S4)  the length are similar:*
$$\mathsf{length}(S1) - \mathsf{length}(S2) = O(\epsilon_3)$$

*with $\epsilon_1, \epsilon_2, \epsilon_3$ being very small. Let $\mathcal{I}_1 = \int_{S_1} \|x - \pi_{12}(x)\|^2 dx$ and $\mathcal{I}_2 = \int_{S_2} \|y - \pi_{21}(y)\|^2 dy$. Then we have*
$$|\mathcal{I}_1 - \mathcal{I}_2| = \sqrt{\mathcal{I}_2}O(\epsilon_1) + \mathcal{I}_2 O(\epsilon_2 + \epsilon_3).$$
*Moreover, if we further assume*

*(S5)  the Hausdorff distance $\mathsf{Haus}(S_1, S_2) = O(\epsilon_4)$ is small,*

*then for any function $\xi: \mathbb{R}^d \mapsto \mathbb{R}$ that has bounded continuous derivative, we have*
$$\int_0^1 \xi(\gamma_1(t))dt = \int_0^1 \xi(\gamma_2(t))dt(1 + O(\epsilon_2 + \epsilon_3 + \epsilon_4)).$$

PROOF. Since $S_1$ and $S_2$ are two bounded, smooth curves. We may parametrized them by $\gamma_1 : [0,1] \mapsto S_1$ and $\gamma_2 : [0,1] \mapsto S_2$ with

$$\begin{aligned} \gamma_1'(t) &= \widetilde{g}_1(\gamma_1(t)), \gamma_1(0) = s_1, \\ \gamma_2'(t) &= \widetilde{g}_2(\gamma_2(t)), \gamma_2(0) = s_2 = \pi_{12}(\gamma_1(0)), \end{aligned} \tag{1}$$

where $\widetilde{g}_1 = \ell_1 g_1$ and $\widetilde{g}_2 = \ell_2 g_2$ for $\ell_1, \ell_2$ being the length of $S_1$ and $S_2$ and $s_1$ one of the end point of $S_1$. The constant $\ell_j$ works as a normalization constant since $g_j$ is an unit vector; it is easy to verify that

$$\mathsf{length}(S_j) = \int_0^1 \|\widetilde{g}_j(t)\| dt = \int_0^1 \ell_j \|g_j(t)\| dt = \ell_j.$$

The starting point $s_2 \in S_2$ must be the projection $\pi_{12}(s_1)$ otherwise the condition (S1) will not hold.

Let

$$\mathcal{I}_1 = \int_0^1 \|\gamma_1(t) - \pi_{12}(\gamma_1(t))\|^2 dt, \quad \mathcal{I}_2 = \int_0^1 \|\gamma_2(t) - \pi_{21}(\gamma_2(t))\|^2 dt. \tag{2}$$

Then the goal is to prove $\mathcal{I}_1 - \mathcal{I}_2 = O(\epsilon_1^2) + O(\epsilon_2^2)$.

Now we consider another parametrization for $S_2$. Let $\eta_2 : [0,1] \mapsto S_2$ such that $\eta_2(t) = \pi_{12}(\gamma_1(t))$. By (S1), $\eta_2$ is a parametrization for $S_2$. The parametrization $\eta_2(t)$ has the following useful properties:

$$\begin{aligned} \eta_2(0) &= \pi_{12}(\gamma_1(0)) = s_2, \\ \eta_2'(t) &= g_2(\eta_2(t)) g_2(\eta_2(t))^T \gamma_1'(t) = g_2(\eta_2(t)) g_2(\pi_{12}(\gamma_1(t)))^T \widetilde{g}_1(\gamma_1(t)). \end{aligned} \tag{3}$$

By condition (S3) and (S4), we have

$$\begin{aligned} g_2(\pi_{12}(\gamma_1(t)))^T \widetilde{g}_1(\gamma_1(t)) &= \ell_1 g_2(\pi_{12}(\gamma_1(t)))^T g_1(\gamma_1(t)) \\ &= \ell_1 (1 + O(\epsilon_2)) \\ &= \ell_2 (1 + O(\epsilon_2) + O(\epsilon_3)) \end{aligned} \tag{4}$$

uniformly for all $t \in [0,1]$. Now apply this result to $\eta_2'(t)$, we obtain that

$$\eta_2'(t) = g_2(\eta_2(t))(1 + O(\epsilon_2) + O(\epsilon_3)). \tag{5}$$

Together with $\eta_2(0) = \gamma_2(0)$, we have

$$\sup_{t \in [0,1]} \|\eta_2(t) - \gamma_2(t)\| = O(\epsilon_2) + O(\epsilon_3). \tag{6}$$

Now by definition of $\mathcal{I}_1$ and the fact that $\pi_{12}^{-1}(\eta_2(t)) = \gamma_1(t)$, we have

$$\begin{aligned} \mathcal{I}_1 &= \int_0^1 \|\gamma_1(t) - \pi_{12}(\gamma_1(t))\|^2 dt \\ &= \int_0^1 \|\pi_{12}^{-1}(\eta_2(t)) - \eta_2(t)\|^2 dt \\ &= \int_0^1 \|\pi_{21}(\eta_2(t)) + O(\epsilon_1) - \eta_2(t)\|^2 dt \quad \text{by (S2)} \\ &= \mathcal{I}_2' + \sqrt{\mathcal{I}_2'} O(\epsilon_1), \end{aligned} \tag{7}$$

where $\mathcal{I}_2' = \int_0^1 \|\pi_{21}(\eta_2(t)) - \eta_2(t)\|^2 dt$.

Now we bound the difference between $\mathcal{I}_2'$ and $\mathcal{I}_2$. Let $U$ be an uniform distribution over $[0,1]$ and define $h(x) : [0,1] \mapsto \mathbb{R}$ as $h(x) = \|\pi_{21}(\gamma_2(x)) - \gamma_2(x)\|$. Note that it is easy to see that $h(x)$ has bounded derivative. Then,

$$\mathcal{I}_2 = \mathbb{E}\|\pi_{21}(\gamma_2(U)) - \gamma_2(U)\|^2 = \mathbb{E}h(U). \tag{8}$$

Since both $\gamma_2$ and $\eta_2$ are parametrization for the curve $S_2$, $\gamma_2^{-1}$ is well defined for all image of $\eta_2$. We define the random variable $W = \gamma_2^{-1}(\eta_2(U))$. Then by definition of $\mathcal{I}_2'$,

$$\mathcal{I}_2' = \mathbb{E}\|\pi_{21}(\eta_2(U)) - \eta_2(U)\|^2 = \mathbb{E}h(W). \tag{9}$$

Since $\sup_{t\in[0,1]}\|\gamma_2'(t) - \eta_2'(t)\| = O(\epsilon_2) + O(\epsilon_3)$, we have $\gamma_2^{-1}(\eta_2(x)) = x + O(\epsilon_2) + O(\epsilon_3)$. Thus, the $p_W(t) - p_U(t) = O(\epsilon_2) + O(\epsilon_3)$, where $p_W$ and $p_U$ are the probability density for random variable $W$ and $U$. Since $U$ is uniform distribution, $p_U = 1$ so that

$$\begin{aligned}
\mathbb{E}h(W) &= \int_0^1 h(t)p_W(t)dt \\
&= \int_0^1 h(t)(p_U(t) + O(\epsilon_2) + O(\epsilon_3))dt \\
&= \int_0^1 h(t)(1 + O(\epsilon_2) + O(\epsilon_3))dt \\
&= \mathbb{E}h(U)(1 + O(\epsilon_2) + O(\epsilon_3)).
\end{aligned} \tag{10}$$

This implies $\mathcal{I}_2' = \mathcal{I}_2(1 + O(\epsilon_2) + O(\epsilon_3))$. Therefore, by (7) we conclude

$$\begin{aligned}
\mathcal{I}_1 &= \mathcal{I}_2' + \sqrt{\mathcal{I}_2'}O(\epsilon_1) \\
&= \mathcal{I}_2 + \sqrt{\mathcal{I}_2}O(\epsilon_1) + \mathcal{I}_2(O(\epsilon_2) + O(\epsilon_3)),
\end{aligned} \tag{11}$$

which completes the proof for the first assertion.

Now we prove the second assertion, here we will assume (S5). Since $\xi$ has bounded first derivative,

$$\begin{aligned}
\int_0^1 \xi(\gamma_1(t))dt &= \int_0^1 \xi(\pi_{12}(\gamma_1(t)))dt(1 + O(\mathsf{Haus}(S_1, S_2))) \\
&= \int_0^1 \xi(\eta_2(t))dt(1 + O(\epsilon_4)).
\end{aligned} \tag{12}$$

Again, let $U$ be the uniform distribution and $W = \gamma_2^{-1}(\eta_2(U))$. We now define the function $\widetilde{h}(t) = \xi(\gamma_2(t))$ for $t \in [0, 1]$. Since both $\xi$ and $\gamma_2$ are bounded differentiable, $\widetilde{h}$ is also bounded differentiable. Then it is easy to see that

$$\begin{aligned}
\int_0^1 \xi(\eta_2(t))dt &= \xi(\gamma_2(t)\gamma_2^{-1}(\eta_2(t)))dt = \mathbb{E}\widetilde{h}(W) \\
\int_0^1 \xi(\gamma_2(t))dt &= \mathbb{E}\widetilde{h}(U).
\end{aligned} \tag{13}$$

Now by the same derivation of (10), we conclude

$$\int_0^1 \xi(\eta_2(t))dt = \mathbb{E}\widetilde{h}(W) = \mathbb{E}\widetilde{h}(U)(1 + O(\epsilon_2) + O(\epsilon_3)). \tag{14}$$

Thus, by (12) and (14), we conclude

$$\int_0^1 \xi(\gamma_1(t))dt = \int_0^1 \xi(\gamma_2(t))dt(1 + O(\epsilon_2) + O(\epsilon_3) + O(\epsilon_4)), \tag{15}$$

which completes the proof.

$\square$

The following Lemma bounds the rate of convergence for the kernel density estimator and will be used frequently in the following derivation.

**Lemma 5 (Lemma 10 of [1]; see also [3])** *Assume (K1–K2) and that* $\log n/n \leq h^d \leq b$ *for some* $0 < b < 1$. *Then we have*

$$||\widehat{p}_n - p||_{k,\max} = O(h^2) + O_P\left(\sqrt{\frac{\log n}{nh^{d+2k}}}\right) \tag{16}$$

*for* $k = 0, \cdots, 3$. *Moreover,*

$$\mathbb{E}||\widehat{p}_n - p||_{k,\max} = O(h^2) + O\left(\sqrt{\frac{\log n}{nh^{d+2k}}}\right). \tag{17}$$

PROOF FOR THEOREM 2.   Here we prove the case for density ridges. The case for density level set can be proved by the similar method. We will use Lemma 4 to obtain the rate. Our strategy is that first we derive $\mathbb{E}(d(U_R, \widehat{R}_n)^2)$ and then show that the other part $\mathbb{E}(d(U_{\widehat{R}_n}, R)^2)$ is similar to the first part.

**Part 1.** We first introduce the concept of reach [2]. For a smooth set $A$, the reach is defined as

$$\mathsf{reach}(A) = \inf\{r : \text{every point in } A \oplus r \text{ has an unique projection onto } A.\}. \tag{18}$$

The reach condition is essential to establish a one-one projection between two smooth sets.

By Lemma 2, property 7 of [1],

$$\mathsf{reach}(R) \geq \min\left\{\frac{\delta_R}{2}, \frac{\beta_2^2}{A_2(\|p^{(3)}\|_{\max} + \|p^{(4)}\|_{\max})}\right\} \tag{19}$$

for some constant $A_2$. Note that $\delta_R$ and $\beta_2$ are the constants in condition (R).

Thus, as long as $\widehat{R}_n$ is close to $R$, every point on $\widehat{R}_n$ has an unique projection onto $R$. Similarly, $\mathsf{reach}(\widehat{R}_n)$ will have a similar bound to $\mathsf{reach}(R)$ whenever $\|\widehat{p}_n - p\|_{4,\max}^*$ is small (reach only depends on fourth derivatives). Hence, every point on $R$ will have an unique projection onto $\widehat{R}_n$. The projections between $R$ and $\widehat{R}_n$ will be one-one and onto except for points near the end points for $R$ and $\widehat{R}_n$. That is, when $\|\widehat{p}_n - p\|_{4,\max}^*$ is sufficiently small, there exists $R^\dagger \subset R$ and $\widehat{R}_n^\dagger \subset \widehat{R}_n$ such that the projection between $R^\dagger$ and $\widehat{R}_n^\dagger$ are one-one and onto. Moreover, the length difference

$$\begin{aligned} \mathsf{length}(R) - \mathsf{length}(R^\dagger) &= O(\mathsf{Haus}(\widehat{R}_n, R)), \\ \mathsf{length}(\widehat{R}_n) - \mathsf{length}(\widehat{R}_n^\dagger) &= O(\mathsf{Haus}(\widehat{R}_n, R)). \end{aligned} \tag{20}$$

Note that by Theorem 6 in [3],

$$\mathsf{Haus}(\widehat{R}_n, R) = O(\|\widehat{p}_n - p\|_{2,\max}^*). \tag{21}$$

Let $x \in R^\dagger$, and let $x' = \pi_{\widehat{R}_n}(x) \in \widehat{R}_n^\dagger$ be its projection onto $\widehat{R}_n$. Then by Theorem 3 in [1] (see their derivation in the proof, the empirical approximation, page 30-32 and equation (79)), we have

$$x' - x = W_2(x)(\widehat{g}_n(x) - g(x))(1 + O(\|\widehat{p}_n - p\|_{3,\max}^*)), \tag{22}$$

where

$$\begin{aligned} W_2(x) &= N(x)H_N^{-1}(x)N(x) \\ H_N(x) &= N(x)^T H(x)N(x) \end{aligned} \tag{23}$$

and $N(x)$ is a $d \times (d-1)$ matrix called the *normal matrix* for $R$ at $x$ whose columns space spanned the normal space for $R$ at $x$. The existence for $N(x)$ is given in Section 3.2 and Lemma 2 in [1]. Thus, we have

$$\mathbb{E}\left(d(x, \widehat{R}_n)^2\right) = \mathbb{E}\left(\|x - x'\|^2\right) = \mathbb{E}\|W_2(x)(\widehat{g}_n(x) - g(x))\|^2 + \Delta_n, \tag{24}$$

where $\Delta_n$ is the remaining term and by Cauchy-Schwartz inequality,

$$\Delta_n \leq \mathbb{E}\left\|W_2(x)(\widehat{g}_n(x) - g(x))\right\|^2 O(\mathbb{E}\|\widehat{p}_n - p\|_{3,\max}^*).$$

Thus,

$$
\begin{aligned}
\mathbb{E}\left(d(x, \widehat{R}_n)^2\right) &= \mathbb{E}\left\|W_2(x)(\widehat{g}_n(x) - g(x))\right\|^2 + \Delta_n \\
&= \mathbb{E}\left\|W_2(x)(\widehat{g}_n(x) - \mathbb{E}(\widehat{g}_n(x)) + \mathbb{E}(\widehat{g}_n(x)) - g(x))\right\|^2 + \Delta_n \\
&= \mathsf{Tr}(\mathsf{Cov}(W_2(x)\widehat{g}_n(x))) + \left\|W_2(x)(\mathbb{E}(\widehat{g}_n(x)) - g(x))\right\|^2 + \Delta_n \\
&= \frac{1}{nh^{d+2}}\mathsf{Tr}(\Sigma(x)) + h^4 b(x)^T b(x) + o\left(\frac{1}{nh^{d+2}}\right) + o\left(h^4\right),
\end{aligned}
\tag{25}
$$

where

$$
\begin{aligned}
\Sigma(x) &= W_2(x)\Sigma(K)W_2(x)p(x), \\
b(x) &= c(K)W_2(x)\nabla(\nabla^2 p(x))
\end{aligned}
\tag{26}
$$

are related to the variance and bias for nonparametric gradient estimation ($\Sigma(K)p(x)$ is the asymptotic covariance matrix for $\widehat{p}_n$ and $c(K)\nabla(\nabla^2 p(x))$ is the asymptotic bias for $\widehat{p}_n$). $\Sigma(K)$ is a matrix and $c(K)$ is a scalar; they both depends only on the kernel function $K$. $\nabla^2 = \frac{\partial^2}{\partial x_1^2} + \cdots + \frac{\partial^2}{\partial x_d^2}$ is the Laplacian operator.

Now we compute $\mathbb{E}(d(U_R, \widehat{R}_n)^2)$. Note that since the length difference between $R$ and $R^\dagger$ is bounded by (20) and (21):

$$
\begin{aligned}
\mathbb{P}(U_R \in R^\dagger) &= 1 - O(\mathbb{E}(\|\widehat{p}_n - p\|_{2,\max}^*)) \\
&= 1 - O(h^2) - O\left(\sqrt{\frac{\log n}{nh^{d+4}}}\right).
\end{aligned}
\tag{27}
$$

Note that we use Lemma 5 to convert the norm into probability bound. By tower property (law of total expectation),

$$
\begin{aligned}
\mathbb{E}(d(U_R, \widehat{R}_n)^2) &= \mathbb{E}(\mathbb{E}(d(U_R, \widehat{R}_n)^2|U_R)) \\
&= \mathbb{E}(\mathbb{E}(d(U_R, \widehat{R}_n)^2|U_R, U_R \in R^\dagger))\mathbb{P}(U_R \in R^\dagger) \\
&\quad + \mathbb{E}(\mathbb{E}(d(U_R, \widehat{R}_n)^2|U_R, U_R \notin R^\dagger))\mathbb{P}(U_R \notin R^\dagger) \\
&= \mathbb{E}\left(\frac{1}{nh^{d+2}}\mathsf{Tr}(\Sigma(U_R)) + h^4 b(U_R)^T b(U_R)\right) + o\left(\frac{1}{nh^{d+2}}\right) + o\left(h^4\right).
\end{aligned}
\tag{28}
$$

Note that by (27), the contribution from $\mathbb{P}(U_R \notin R^\dagger)$ is smaller than the main effect in (25) so we absorb it into the small $o$ terms. Defining $B_R^2 = \mathbb{E}(b(U_R)^T b(U_R))$ and $\sigma_R^2 = \mathbb{E}(\mathsf{Tr}(\Sigma(U_R)))$, we obtain

$$\mathbb{E}(d(U_R, \widehat{R}_n)^2) = B_R^2 h^4 + \frac{\sigma_R^2}{nh^{d+2}} + o\left(\frac{1}{nh^{d+2}}\right) + o\left(h^4\right).
\tag{29}$$

**Part 2.** We have proved the first part for the $\mathcal{L}_2$ coverage risk. Now we prove the result for $\mathbb{E}(d(U_{\widehat{R}_n}, R)^2)$; this will apply Lemma 4. If we think of $R^\dagger$ as $S_1$ and $\widehat{R}_n^\dagger$ as $S_2$ in Lemma 4, then

$$
\begin{aligned}
\mathbb{E}(d(U_{R^\dagger}, \widehat{R}_n^\dagger)^2|X_1, \cdots, X_n) &= \int_0^1 \|\gamma_1(t) - \pi_{12}(\gamma_1(t))\|^2 dt = \mathcal{I}_1 \\
\mathbb{E}(d(U_{\widehat{R}_n^\dagger}, R^\dagger)^2|X_1, \cdots, X_n) &= \int_0^1 \|\gamma_2(t) - \pi_{21}(\gamma_2(t))\|^2 dt = \mathcal{I}_2.
\end{aligned}
\tag{30}
$$

Thus, $\mathbb{E}(d(U_{\widehat{R}_n^\dagger}, R^\dagger)^2)$ is approximated by $\mathbb{E}(d(U_{R^\dagger}, \widehat{R}_n^\dagger)^2)$ if the $\epsilon_1, \epsilon_2, \epsilon_3$ in Lemma 4 is small. Here we bound $\epsilon_j$.

The bound for $\epsilon_1$ is simple. For all $x \in S_1$, let $\theta$ be the angle between the two vectors $v_1 = \pi_{12}(x) - x$ and $v_2 = \pi_{21}^{-1}(x) - x$. By the property of projection, $v_1$ is normal to $\widehat{R}_n$ at $\pi_{12}(x)$

and $v_2$ is normal to $R$ at $x$. Thus, by Lemma 2 properties 5 and 6 of [1], the angle $\theta$ is bounded by $O(\|\widehat{p}_n - p\|^*_{3,\max})$. Note that their Lemma proves the normal matrices $N(x)$ and $\widehat{N}_n(\pi_{12}(x))$ are close which implies the canonical angle between two subspace are close so that $\theta$ is bounded. Now by the fact that both $\|\pi_{12}(x) - x\|$ and $\|\pi_{21}^{-1}(x) - x\|$ are bounded by $\mathsf{Haus}(\widehat{R}_n, R)$, we conclude $\epsilon_1 \leq \mathsf{Haus}(\widehat{R}_n, R) \times \theta = O(\|\widehat{p}_n - p\|^{*2}_{3,\max})$.

For $\epsilon_2$, we will use the property of normal matrix $N(x)$. Let $\widehat{N}_n(x)$ be the normal matrix for $\widehat{R}_n$ at $x$. By Lemma 2, properties 5 and 6 of [1],

$$\|N(x)N(x)^T - \widehat{N}_n(\pi_{\widehat{R}_n}(x))\widehat{N}_n(\pi_{\widehat{R}_n}(x))^T\|_{\max} = O(\mathsf{Haus}(\widehat{R}_n, R)) + O(\|\widehat{p}_n - p\|^*_{3,\max})$$
$$= O(\|\widehat{p}_n - p\|^*_{3,\max}).$$

$N(x)N(x)^T$ is the projection matrix onto normal space; so the tangent vector is perpendicular to that projection. The bounds for the two projection matrix implies the bound to the two tangent vectors. Thus, $\epsilon_2 = O(\|\widehat{p}_n - p\|^*_{3,\max})$.

For $\epsilon_3$, since the smoothness for $\widehat{R}_n$ is similar to $R$ (the normal direction is similar by $\epsilon_2$) and their Hausdorff distance is bounded by $O(\|\widehat{p}_n - p\|^*_{2,\max})$. The length difference is at the same rate of Hausdorff distance. Thus, we may pick $\epsilon_3 = O(\|\widehat{p}_n - p\|^*_{2,\max})$.

Let $\mathcal{I}_1 = \mathbb{E}(d(U_{R^\dagger}, \widehat{R}_n^\dagger)^2 | X_1, \cdots, X_n)$ and $\mathcal{I}_2 = \mathbb{E}(d(U_{\widehat{R}_n^\dagger}, R^\dagger)^2 | X_1, \cdots, X_n)$. By Lemma 4 and the above choice for $\epsilon_j$, we conclude

$$\mathcal{I}_1 = \mathcal{I}_2(1 + O(\|\widehat{p}_n - p\|^*_{3,\max})) + \sqrt{\mathcal{I}_2}O(\|\widehat{p}_n - p\|^{*2}_{3,\max}). \tag{31}$$

Thus, by tower property again (taking expectation over both side) and Lemma 5 $\mathbb{E}\|\widehat{p}_n - p\|^*_{3,\max} = O(h^2) + O\left(\sqrt{\frac{\log n}{nh^{d+6}}}\right) = o(1)$,

$$\mathbb{E}(d(U_{R^\dagger}, \widehat{R}_n^\dagger)^2) = \mathbb{E}(\mathcal{I}_1) = \mathbb{E}(\mathcal{I}_2) + o(1) = \mathbb{E}(d(U_{\widehat{R}_n^\dagger}, R^\dagger)^2) + o(1). \tag{32}$$

Now since by (20) and the fact that $\mathbb{E}\mathsf{Haus}(\widehat{R}_n, R) = o(1)$, we have

$$\mathbb{E}(d(U_{R^\dagger}, \widehat{R}_n^\dagger)^2) = \mathbb{E}(d(U_R, \widehat{R}_n)^2)(1 + o(1))$$
$$\mathbb{E}(d(U_{\widehat{R}_n^\dagger}, R^\dagger)^2) = \mathbb{E}(d(U_{\widehat{R}_n}, R)^2)(1 + o(1)). \tag{33}$$

Combining by (29), (32) and (33), we conclude

$$\mathsf{Risk}_{2,n} = \frac{\mathbb{E}(d(U_R, \widehat{R}_n)^2) + \mathbb{E}(d(U_{\widehat{R}_n}, R)^2)}{2}$$
$$= \mathbb{E}(d(U_R, \widehat{R}_n)^2) + o(1) \tag{34}$$
$$= B_R^2 h^4 + \frac{\sigma_R^2}{nh^{d+2}} + o\left(\frac{1}{nh^{d+2}}\right) + o\left(h^4\right),$$

where $B_R^2 = \mathbb{E}(b(U_R)^T b(U_R))$ and $\sigma_R^2 = \mathbb{E}(\mathsf{Tr}(\Sigma(U_R)))$. Note that all the above derivation works only when

$$\mathbb{E}\|\widehat{p}_n - p\|^*_{3,\max} = O(h^2) + O\left(\sqrt{\frac{\log n}{nh^{d+6}}}\right) = o(1). \tag{35}$$

This requires $h \to 0$ and $\frac{\log n}{nh^{d+6}} \to 0$, which constitutes the conditions on $h$ we need.

$\square$

PROOF FOR THEOREM 3. Since we are proving the bootstrap consistency, we assume $X_1, \cdots, X_n$ are given.

By Theorem 2, the estimated risk $\widehat{\mathsf{Risk}}_{n,2}$ has the following asymptotic behavior

$$\widehat{\mathsf{Risk}}_{n,2} = \widehat{B}_R^2 h^4 + \frac{\widehat{\sigma}_R^2}{nh^{d+2}} + o\left(\frac{1}{nh^{d+2}}\right) + o\left(h^4\right), \tag{36}$$

where

$$\begin{aligned}
\widehat{B}_R^2 &= \mathbb{E}\left(\widehat{b}_n(U_{\widehat{R}_n})^T \widehat{b}_n(U_{\widehat{R}_n})|X_1,\cdots,X_n\right), \\
\widehat{\sigma}_R^2 &= \mathbb{E}\left(\mathsf{Tr}(\widehat{\Sigma}_n(U_{\widehat{R}_n}))|X_1,\cdots,X_n\right)
\end{aligned} \tag{37}$$

with $\widehat{b}_n(x) = c(K)W_2(x)\nabla(\nabla^2\widehat{p}_n(x))$ and $\widehat{\Sigma}_n(x) = W_2(x)\Sigma(K)W_2(x)\widehat{p}_n(x)$ from (26). To prove the bootstrap consistency, it is equivalent to prove that $\widehat{B}_R^2$ and $\widehat{\sigma}_R^2$ converges to $B_R$ and $\sigma_R^2$.

Here we prove the consistency for $\widehat{B}_R$. The consistency for $\widehat{\sigma}_R$ can be proved in the similar way. We define the following two functions

$$\begin{aligned}
\widehat{\Omega}_n(x) &= \|c(K)W_2(x)\nabla(\nabla^2\widehat{p}_n(x))\|^2, \\
\Omega(x) &= \|c(K)W_2(x)\nabla(\nabla^2 p(x))\|^2.
\end{aligned} \tag{38}$$

It is easy to see that $\widehat{B}_R^2 = \mathbb{E}\left(\widehat{\Omega}_n(U_{\widehat{R}_n})|X_1,\cdots,X_n\right)$ and $B_R^2 = \mathbb{E}\left(\Omega(U_R)\right)$.

Similarly as in the proof for Theorem 2, we define $\widehat{R}_n^\dagger \subset \widehat{R}_n$ that has one-one and onto projection to $R^\dagger$. By (27), we can replace $U_{\widehat{R}_n}$ by $U_{\widehat{R}_n^\dagger}$ and $U_R$ by $U_{R^\dagger}$ at the cost of probability $O(h^2) + O\left(\sqrt{\frac{\log n}{nh^{d+4}}}\right)$.

Now we will apply Lemma 4 again to prove the result. Again, we think of $R^\dagger$ as $S_1$ and $\widehat{R}_n^\dagger$ as $S_2$. Let $U$ be an uniform distribution over $[0,1]$. Then the random variable $U_{R^\dagger} = \gamma_1(U)$ and $U_{\widehat{R}_n^\dagger} = \gamma_2(U)$. Thus,

$$\mathbb{E}\left(\Omega(U_{R^\dagger})\right) = \int_0^1 \Omega(\gamma_1(t))dt, \quad \mathbb{E}\left(\widehat{\Omega}_n(U_{\widehat{R}_n^\dagger})|X_1,\cdots,X_n\right) = \int_0^1 \widehat{\Omega}_n(\gamma_2(t))dt. \tag{39}$$

By the second assertion in Lemma 4,

$$\begin{aligned}
\mathbb{E}\left(\Omega(U_{R^\dagger})\right) &= \int_0^1 \Omega(\gamma_1(t))dt \\
&= \int_0^1 \Omega(\gamma_2(t))dt(1 + O(\epsilon_2) + O(\epsilon_3) + O(\epsilon_4)) \\
&= \int_0^1 \Omega(\gamma_2(t))dt(1 + O(\|\widehat{p}_n - p\|_{3,\max}^*)).
\end{aligned} \tag{40}$$

Note that we use the fact that $\mathsf{Haus}(\widehat{R}_n, R) = O(\|\widehat{p}_n - p\|_{2,\max}^*)$. Since $\Omega$ only involves third derivative for the density $p$, we have $\sup_{x\in\mathbb{R}^d}\|\Omega(x) - \widehat{\Omega}_n(x)\| = O(\|\widehat{p}_n - p\|_{3,\max})$. This implies

$$\int_0^1 \Omega(\gamma_2(t))dt = \int_0^1 \widehat{\Omega}_n(\gamma_2(t))dt + O(\|\widehat{p}_n - p\|_{3,\max}). \tag{41}$$

Now combining all the above and the definition for $\widehat{B}_R$, we conclude

$$
\begin{aligned}
\widehat{B}_R^2 &= \mathbb{E}\left(\widehat{\Omega}_n(U_{\widehat{R}_n})|X_1,\cdots,X_n\right) \\
&= \mathbb{E}\left(\widehat{\Omega}_n(U_{\widehat{R}_n^\dagger})|X_1,\cdots,X_n\right) + O(\mathsf{Haus}(\widehat{R}_n,R)) \\
&= \int_0^1 \widehat{\Omega}_n(\gamma_2(t))dt + O(\mathsf{Haus}(\widehat{R}_n,R)) \quad \text{(by (39))} \\
&= \int_0^1 \Omega(\gamma_2(t))dt + O(\|\widehat{p}_n - p\|_{3,\max}) \quad \text{(by (41))} \\
&= \mathbb{E}\left(\Omega(U_{R^\dagger})\right) + O(\|\widehat{p}_n - p\|_{3,\max}) \quad \text{(by (40))} \\
&= \mathbb{E}\left(\Omega(U_R)\right) + O(\|\widehat{p}_n - p\|_{3,\max}) \\
&= B_R^2 + O(\|\widehat{p}_n - p\|_{3,\max}).
\end{aligned}
\tag{42}
$$

Therefore, as along as we have $\|\widehat{p}_n - p\|_{3,\max} = o_P(1)$, we have

$$
\widehat{B}_R^2 - B_R^2 = o_P(1).
\tag{43}
$$

Similarly, we the same condition implies

$$
\widehat{\sigma}_R^2 - \sigma_R^2 = o_P(1).
\tag{44}
$$

Now recall from (36) and Theorem 2, the risk difference is

$$
\begin{aligned}
\widehat{\mathsf{Risk}}_{n,2} - \mathsf{Risk}_{n,2} &= (\widehat{B}_R^2 - B_R^2)h^4 + \frac{\widehat{\sigma}_R^2 - \sigma_R^2}{nh^{d+2}} + o\left(h^4\right) + o\left(\frac{1}{nh^{d+2}}\right) \\
&= o_P\left(h^4\right) + o_P\left(\frac{1}{nh^{d+2}}\right) \quad \text{(by (43) and (44))}.
\end{aligned}
\tag{45}
$$

Since Theorem 2 implies $\mathsf{Risk}_{n,2} = O\left(h^4\right) + O\left(\frac{1}{nh^{d+2}}\right)$, by (45) we have

$$
\frac{\widehat{\mathsf{Risk}}_{n,2} - \mathsf{Risk}_{n,2}}{\mathsf{Risk}_{n,2}} = o_P(1)
\tag{46}
$$

which proves the theorem.

Note that in order (46) to hold, we need $\|\widehat{p}_n - p\|_{3,\max} = o_P(1)$. By Lemma 5,

$$
\|\widehat{p}_n - p\|_{3,\max} = O(h^2) + O_P\left(\sqrt{\frac{\log n}{nh^{d+6}}}\right).
\tag{47}
$$

Thus, a sufficient condition to $\|\widehat{p}_n - p\|_{3,\max} = o_P(1)$ is to pick $h$ such that $\frac{\log n}{nh^{d+6}} \to 0$ and $h \to 0$. This gives the restriction for the smoothing parameter $h$.

$\square$

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

—