[Reviews · NeurIPS 2015]

Submitted by Assigned_Reviewer_1

I like the paper; to my knowledge the proposed technique is novel, and while it's pretty straightforward, it would be good to publish. The paper is well written.

The main concern I have is related to the practical choice of estimator. The authors describe smoothed bootstrap before data splitting; the results, e.g. Theorem 3, are for bootstrap (with the result for data splitting mentioned as having similar derivation). Yet, in all of the experiments only data splitting method appears to be used. Why? If the authors believe that the bootstrap method is valid, it should be evaluated empirically. If not, it should be dropped from the paper altogether.

Some minor points and questions: - l. 126: M, M-hat should be R and R-hat, I believe - l. 158: "The we" --> "Then we" - l. 213: "is take "-> "is taken"

- Fig.2: what are the dashed vs. solid lines depicting in the right subfig? I could not find anything in either the caption or the text that would explain those.

- Fig.3: Why are some of the plots on the left, especially the middle one (spirals) so non-monotonic? My intuition is that the typical curve in such plot should have a U shape: going from undersmoothing to (about) right to oversmoothing. I understand that some noise will afect the curve in case of finite data. Still, the second drop in the spirals plot is striking (the drops in the other two plots for high smoothing regimes are more plausibly explained away by noise). Can the authors offer some comment on this?
Summary: The paper proposes what I believe is a novel approach to density ridge detection; the main innovation is the definition of risk based on Hausdorff set distance measure between predicted and true ridges. Practically, ridge detection is done by subspace constrained mean shift which relies on a kernel density estimator; the choice of bandwidth for that KDE is where the novel risk definition plays its role (the bandwidth is chosen to minimize this risk, as estimated either by cross-validation or by smoothed bootstrap). Theoretical results include consistency proofs; empirical evaluation is mostly on synthetic data, with some anecdotal real data examples.

Submitted by Assigned_Reviewer_2

This paper considers the density ridge estimation problem and introduces a new concept coverage risk as an error measurement. This is a potentially interesting problem. However, I found the current form the paper is not acceptable for NIPS.

1. The paper is not easy to read. In particular, the authors did not spend enough effort to motivate the problem. For example, why the coverage risk is important?

2. I did not quite catch the point of Section 3.

3. Why it's called optimal ridge detection? In what sense it is optimal?

4. Many notations are not very clear. For example, on Page 5, the authors wrote BC^3. I guess this means the class of functions whose 3rd order derivatives are bounded and continuous? It would be important to make these notation clearer.

Summary: This paper seems to be written in a very rush and sloppy way. Though the problem being considered is very interesting. The authors did not put enough effort to make the essential results deliverable to the audience.

Submitted by Assigned_Reviewer_3

The paper focuses on density ridge estimation. The authors propose the use of coverage risk which is a generalization of the mean integrated expected error. The authors propose two estimators (based on either smoothed bootstrap or data splitting), prove their consistency, and study convergence rates. Particularly nice is that, unlike some of the cited works, the proposed method does not suffer from the problems related to the notorious estimation of the smoothing parameter.

Experiments are done on simulated data and Cosmic Web data.

Quality: The paper is technically sound, and its overall quality is high. The problem attacked is interesting and of importance.

Clarity: The paper is well written and clear.

Originality: As far as I am aware of, the idea presented is new.

Significance: I believe this work will have a nice impact.

A remark on the experiments: In both Figures 3 and 4, the authors show under-smoothing and over-smoothing results next to theirs. That's fine, but what is really missing here is a comparison with what ones gets when one uses simple cross validation for the optimal smoothing parameter for standard Kernel Density Estimation (e.g., using a Gaussian kernel) and then does the ridge extraction after the fact. Presumably, according to the authors' claims this would be inferior to their method which finds the optimal smoothing parameter wrt the task at hand (ridge detection) as opposed to the optimal parameter for general-purpose KDE. Such an experimental result, whose absence is disappointing, would have strengthened the authors' main claim about the utility of the method and would have also convinced me to give their paper a higher grade.

A remark regarding principal curves:

The authors may want to also cite Hastie's work from the 80's.

----------------------------- ----------------------------- Having read the rebuttal, my impression remained positive. I keep my score as it is. That said, I am sightly concerned about the wording used in the rebuttal regarding my remark on cross validation: while they committed themselves in the rebuttal to add the results obtained using ordinary CV, they avoided stating whether this comparison indeed makes their method look favorable. I hope this is just a poor choice of wording and nothing more. It will be very disappointing to find out that the results are actually the same.

Summary: I like this paper. It is well written, attacks the problem in a principled way, provides a decent theoretical analysis, and the method seems to be effective with a potential nice impact.

Author Feedback
Author rebuttal: Reviewer 1:

We included the smoothed bootstrap for completeness.
In practice the splitting method worked much better.
We will add a remark about this.
(We can remove it if you prefer).

Minor comments: we will fix all of these.

Fig 2. Dashed line is coverage from data points (black dots) over green/orange curves in the left panel and solid line is coverage from green/orange curves on data points.
We will add this.

Fig 3. The second min in middle row corresponds to a phase transition when the estimator becomes a big circle-this is also a locally stable result.
We will add a remark about this.

Reviewer 2:

We are surprised you found the paper hard to read.
(The other two main reviewers liked the paper and rate it as 8.)

1. The coverage risk is important for the same reason that mean squared error is important for estimating a regression function.
It is simply a measure of error.
Minimizing it gives us a way to choose the tuning parameter.
We will clarify the motivation for the approach in the introduction.

2. We will rewrite the intro (and title) of Section 3.
The key point is that we can use our risk as a way to measure the distance between two manifolds.

3. We called it optimal because we are minimizing the coverage risk.

4. You are correct: BC^3 is the class of functions with three bounded continuous
derivatives.
We will clarify the notation.

Reviewer 3:

We will show the results we obtain using ordinary cross-validation, as you suggest.

We will add the cites to Hastie et al.

Reviewer 4:

There is no practical difference between the L1 and L2 risk.
The Hausdorff distance does not work as well.
We can add a comment about this.

Reviewer 5:

We're not sure why you rated the paper so low.
Two of the main reviewers rated the paper as 8.
We hope you will have a closer look at the paper and reconsider.